# Comparison of Clinical Characteristics and Outcomes of Younger and Elderly Patients with Severe COVID-19 in Korea: A Retrospective Multicenter Study

**DOI:** 10.3390/jpm11121258

**Published:** 2021-11-29

**Authors:** Gil Myeong Seong, Ae-Rin Baek, Moon Seong Baek, Won-Young Kim, Jin Hyoung Kim, Bo Young Lee, Yong Sub Na, Song-I Lee

**Affiliations:** 1Department of Internal Medicine, Jeju National University, Jeju 63243, Korea; rolland0211@gmail.com; 2Department of Internal Medicine, Division of Allergy and Pulmonology, Soonchunhyang University Bucheon Hospital, Bucheon 14584, Korea; aerinbaek@gmail.com; 3Department of Internal Medicine, College of Medicine, Chung-Ang University, Seoul 06973, Korea; wido21@cau.ac.kr (M.S.B.); wykim81@cau.ac.kr (W.-Y.K.); 4Department of Internal Medicine, Division of Respiratory and Critical Care Medicine, Ulsan University Hospital, College of Medicine, University of Ulsan, Ulsan 44033, Korea; firebomb@daum.net; 5Division of Allergy and Respiratory Diseases, Soonchunhyang University Seoul Hospital, Seoul 04401, Korea; etboss2@gmail.com; 6Department of Pulmonology and Critical Care Medicine, Chosun University Hospital, Gwangju 61453, Korea; ebusters@chosun.ac.kr; 7Department of Pulmonary and Critical Care Medicine, Chungnam National University Hospital, Daejeon 35015, Korea

**Keywords:** aged, COVID-19, elderly

## Abstract

Old age is associated with disease severity and poor prognosis among coronavirus disease 2019 (COVID-19) cases; however, characteristics of elderly patients with severe COVID-19 are limited. We aimed to assess the clinical characteristics and outcomes of patients hospitalized with severe COVID-19 at tertiary care centers in South Korea. This retrospective multicenter study included patients with severe COVID-19 who were admitted at seven hospitals in South Korea from 2 February 2020 to 28 February 2021. The Cox regression analyses were performed to assess factors associated with the in-hospital mortality. Of 488 patients with severe COVID-19, 318 (65.2%) were elderly (≥65 years). The older patient group had more underlying diseases and a higher severity score than the younger patient group. The older patient group had a higher in-hospital mortality rate than the younger patient group (25.5% versus 4.7%, *p*-value < 0.001). The in-hospital mortality risk factors among patients with severe COVID-19 included age, acute physiology and chronic health evaluation II score, presence of diabetes and chronic obstructive lung disease, high white blood cell count, low neutrophil-lymphocyte ratio and platelet count, do-not-resuscitate order, and treatment with invasive mechanical ventilation. In addition to old age, disease severity and examination results must be considered in treatment decision-making.

## 1. Introduction

Coronavirus disease 2019 (COVID-19) is primarily caused by severe acute respiratory syndrome coronavirus 2 (SARS-CoV-2), which is transmitted from person to person via close contact with an infected individual. A pandemic outbreak was declared by the World Health Organization (WHO) in March 2020 [1]. In South Korea, from 3 January 2020 to 15 November 2021, there have been 397,466 confirmed cases of COVID-19, with 3115 deaths, reported to WHO. As of 6 November 2021, a total of 79,356,440 vaccine doses have been administered [2]. Vaccination [3,4,5] is currently in place worldwide and this is expected to reduce the incidence of COVID-19, but the world is still struggling with COVID-19.

The most common underlying diseases as predictors of mortality among COVID-19 patients were hypertension, diabetes, and cardiovascular diseases [6]. In addition, PaO_2_/FiO_2_ ratio (*p*/F ratio) ≤ 200 mmHg, respiratory failure at admission, high neutrophil and low lymphocyte, platelet, and albumin levels, and high sequential organ failure assessment (SOFA) score are known to be associated with mortality [7,8,9]. Old age is associated with greater disease severity and poor prognosis among patients with COVID-19 [7,8,9,10].

Elderly patients with COVID-19 present with more atypical symptoms and commonly experience progression to severe COVID-19 compared with younger patients [11,12]. In addition, they have a higher in-hospital mortality rate and a longer length of hospital stay [13,14]. The higher risk of COVID-19 mortality among older people occurs because older people are more likely to have other risk factors such as underlying disease and a high frailty scale. Also, older age is independently associated with COVID-19 mortality without other risk factors [15,16]. However, data regarding the prognostic factors and characteristics of elderly patients with severe COVID-19 in a rapidly aging society are limited. Therefore, the current study aimed to assess the clinical characteristics and outcomes of patients hospitalized with severe COVID-19 at tertiary care centers in South Korea.

## 2. Materials and Methods

This multicenter retrospective study analyzed data collected from all patients with severe COVID-19 who were hospitalized at seven hospitals in Korea from 2 February 2020 to 28 February 2021. The research was approved by the ethical committee of the promoting center (institutional review board of Chungnam National University Hospital, approval no. 2021-04-053, approval date: 30 April 2021) and by the local ethical committees. The need for a written informed consent was waived because of the retrospective nature of the study. During the study period, 1565 patients were screened (Figure 1). A total of 1077 patients were excluded due to non-severe COVID-19.

All data were retrieved from the electronic medical records. Information about the characteristics of patients, including sex, age, symptoms, and initial laboratory and radiologic findings, was collected. Moreover, data about the need for invasive treatment (mechanical ventilation, treatment with vasopressor, and continuous renal replacement therapy (CRRT), in-hospital mortality, and duration of hospital stay were extracted. The initial Acute Physiology and Chronic Health Evaluation II (APACHE II) score, Charlson Comorbidity Index (CCI), and Clinical Frailty Scale (CFS) score were analyzed to evaluate the patient’s condition upon admission.

### 2.1. Definition

The approval of requests was reserved for hospitalized patients who had SARS-CoV-2 infection confirmed via reverse-transcriptase polymerase-chain-reaction (PCR). Severe COVID-19 infection was defined as including an oxygen saturation level of 94% or less while the patient was breathing ambient air, and a need for oxygen support [17]. Secondary infection was defined as the presence of clinical signs and/or symptoms of infection and the presence of a pathogen based on the diagnostic tests, including respiratory bacterial PCR (using endotracheal aspirates and expectorated sputum), nasopharyngeal PCR, blood culture, and urine culture during illness or hospital stay [18].

### 2.2. Statistical Analysis

All values were expressed as median (interquartile range (IQR): 25–75 percentile) for continuous variables or as percentages for categorical variables. The student’s t-test or the Mann–Whitney U test was used for continuous data, and the Pearson’s chi-squared test or the Fisher’s exact test for categorical data. The risk factors for mortality were analyzed using the Cox proportional hazards model with backward, stepwise elimination, and variables with *p* < 0.1 in the univariate analysis were entered into the multivariate models. All *p*-values were two-tailed, and a *p* value of <0.05 was considered statistically significant. All statistical analyses were performed using the Statistical Package for the Social Sciences software (version 22.0; IBM Corporation, Somers, NY, USA).

## 3. Results

### 3.1. Baseline Characteristics of Patients with Severe COVID-19

Of the total 1565 COVID-19 patients hospitalized, 488 who had severe COVID-19 were included in this study. Moreover, 170 (34.8%) and 318 (65.2%) patients were aged <65 years (younger patient group) and ≥65 years (older patient group), respectively (Figure 1).

Table 1 shows the baseline characteristics of patients. The median ages of younger and older patient groups were 58 (IQR: 51–62) years and 78 (IQR: 71–84) years, respectively. The older patient group had a lower proportion of male patients and fewer symptoms upon admission. Further, the older patient group had a lower body mass index (BMI) and a higher proportion of nursing facility residents. Older patients had higher APACHE II scores (11.5 (9.0–15.0) versus. 7.0 (5.0–10.0), *p* < 0.001), CCI (4.0 (3.0–5.0) versus 2.0 (1.0–2.0), *p* < 0.001), and CFS scores (4.0 (3.0–7.0) versus 2.0 (1.0–3.0), *p* < 0.001) than younger patients. Hypertension, diabetes mellitus (DM), chronic obstructive pulmonary disease (COPD), cardiovascular disease, and heart failure were more common in older patients than in younger patients. In terms of vital signs, the older patients had a lower diastolic blood pressure (DBP), heart rate (HR), and body temperature than the younger patients. Regarding the initial laboratory data, compared with the younger patients, the older patients had lower neutrophil-to-lymphocyte ratios (NLRs), hemoglobin levels, and albumin levels, and higher blood urea nitrogen (BUN) levels. Based on the initial radiography result, unilateral lesions were more common in older patients than in younger patients.

### 3.2. Treatment and Clinical Outcomes of Patients with Severe COVID-19

Table 2 shows the treatment and outcomes of patients with severe COVID-19. The use of antibiotics (66.4% versus 48.8%, *p* < 0.001) and vasopressors (15.1% versus 7.6%, *p* = 0.018) was higher in older patients than in younger patients. In terms of the oxygen delivery system, nasal prongs (88.8% versus 80.8%, *p* = 0.023) were more commonly used in younger patients. The overall in-hospital mortality rate was 18.2% (89 of 488), and the older group had a higher mortality rate than the younger group (25.5% versus 4.7%, *p* < 0.001). The older patient group had a higher incidence of secondary infection (32.4% versus 23.5%, *p* = 0.040) and a higher proportion of patients with a do-not-resuscitate (DNR) order (23.9% versus 2.9%, *p* < 0.001) than the younger patient group.

### 3.3. Factors Associated with In-Hospital Mortality

Table 3 shows the results of the multivariate analysis of factors associated with in-hospital mortality. After adjusting for confounders, the independent predictors of in-hospital mortality included age (hazard ratio (HR): 1.031, 95% confidence interval (CI): 1.008–1.054; *p* = 0.008), APACHE II score (HR: 1.038, 95% CI: 1.006–1.071; *p* = 0.018), presence of DM (HR: 1.622, 95% CI: 1.029–2.559; *p* = 0.037) and COPD (HR: 4.294, 95% CI: 1.874–9.837; *p* = 0.001), high white blood cell (WBC) count (HR: 1.104, 95% CI: 1.056–1.153; *p* < 0.001), low NLR (HR: 0.984, 95% CI: 0.968–1.000; *p* = 0.044), low platelet count (HR: 0.996, 95% CI: 0.993–0.999; *p* = 0.024), DNR order (HR: 6.260, 95% CI: 3.792–10.333; *p* < 0.001), and treatment with invasive mechanical ventilation (HR: 1.824, 95% CI: 1.070–3.107; *p* = 0.027).

## 4. Discussion

This multicenter study investigated the correlation between in-hospital mortality and some factors among elderly patients with severe COVID-19. Approximately 65.2% of hospitalized patients with severe COVID-19 were aged over 65 years. Older patients had higher initial APACHE II, CCI, and CFS scores than younger patients. Moreover, they commonly presented with underlying diseases, lower NLRs and hemoglobin and albumin levels, and higher BUN levels. The use of antibiotics and vasopressors was higher in older patients than in younger patients. Older patients had a higher in-hospital mortality rate and incidence of secondary infection than younger patients, and DNR order were more common in older patients than in younger patients. The risk factors for in-hospital mortality in patients with severe COVID-19 included high APACHE II score and WBC count, low platelet count, DNR order, and treatment with invasive mechanical ventilation. The in-hospital mortality rate of the older patient group was 25.5%.

There is a large population of older patients with severe COVID-19. Tan et al. showed that 47.2% of patients who died from COVID-19 were elderly (≥70 years) [19]. Singhal et al. revealed that 51% of older patients with COVID-19 (≥60 years) had severe infection and 22% of patients were critically ill [20]. In the study by Agnieszka et al., 50% of elderly patients with COVID-19 (≥60 years) were hospitalized (50.5%), of whom 23.5% were admitted at the ICU [21]. Although several studies have reported the proportion of elderly patients with COVID-19, different age criteria were used, and patients of varying severity were included. Nevertheless, they showed that elderly patients account for a large proportion of patients with severe or critical COVID-19. This result is similar to that of our research.

This study showed that hypertension was the most common comorbidity in this study, and elderly patients frequently presented with DM, COPD, cardiovascular disease, and heart failure. These results are similar to those of other studies showing that patients commonly had hypertension, and diabetes, cardiovascular disease, hypercholesterolemia, chronic lung disease, and malignancy were the frequent comorbidities [20,21,22,23,24]. In this research, elderly patients had lower NLRs, hemoglobin and albumin levels, and higher BUN levels. Other studies showed that patients presented with anemia, lymphopenia, thrombocytopenia, slightly abnormal creatinine and BUN levels, low albumin levels, high d-dimer, CRP, and procalcitonin levels [20,21,22,23,24,25]. Although the findings slightly differed, this study only included patients with severe COVID-19, and this might have influenced the results. Nevertheless, most laboratory data trends were similar.

In this study, the risk factors for in-hospital mortality among patients with severe COVID-19 included age, higher APACHE II score, presence of DM and COPD, higher WBC count, lower NLR, platelet count, DNR order, and treatment with invasive mechanical ventilation. Based on the study of Grasselli, which included patients with COVID-19 admitted to the ICU, the independent risk factors associated with mortality included old age (HR: 1.75, 95% CI: 1.60–1.92), male sex (HR: 1.57, 95% CI: 1.31–1.88), high fraction of inspired oxygen (HR: 1.14, 95% CI: 1.10–1.19), or low partial pressure of oxygen-to-fraction of inspired oxygen ratio (HR: 0.80, 95% CI: 0.74–0.87) upon ICU admission, and history of COPD (HR: 1.68, 95% CI: 1.28–2.19), hypercholesterolemia (HR: 1.25, 95% CI: 1.02–1.52), and type 2 diabetes (HR: 1.18, 95% CI: 1.01–1.39) [26]. According to Kim et al., the independent factors associated with in-hospital mortality among hospitalized patients with COVID-19 were old age, immunosuppression, renal, chronic lung, cardiovascular, and neurologic disorders, and diabetes [27]. Other studies have shown that patient severity and malnutrition could affect patient prognosis [28,29]. Therefore, higher severity scores, malnutrition, poor laboratory data, and DNR orders were associated with in-hospital mortality even in the patient group with severe COVID-19.

Elderly patients with severe COVID-19 had higher severity and Clinical Frailty Scale scores, and they commonly presented with underlying diseases such as DM and COPD. These results are similar to those of other studies [30,31,32,33,34,35]. In Guo et al.'s study, elderly patients had more underlying diseases, and common comorbidities included hypertension, diabetes, and cardiac disease [25]. In the study by Gao et al., which included elderly patients with COVID-19 (≥65 years), the common morbidities included cardiovascular diseases (49% versus 20%), respiratory diseases (51% versus 11%), and chronic kidney disease (29% versus 5%) in the deceased group in elderly COVID-19 patients (aged ≥65 years), and cerebrovascular disease than in the discharged group. In addition, high CRP and BUN level and lymphopenia were associated with poor prognosis [35]. Similar to our study, several studies showed that underlying disease was more common and mortality rate was higher in the elderly group.

This study had several limitations. First, it included patients admitted at tertiary or referral hospitals capable of critical care. This might have affected the results, as patients who had been transferred from other hospitals or from living treatment centers due to worsening conditions were included. Second, instead of manually reviewing medical records, data were collected from the electronic health record database. Hence, some details could have been missing. Third, patients’ economic status was not assessed even though inferior economic conditions are associated with poor outcomes [36,37]. Nevertheless, in Korea, the National Health Insurance provides free COVID-19 treatment. Hence the impact is likely low. Fourth, in South Korea, at the beginning of the COVID-19 outbreak, regardless of severity, all patients were hospitalized and then discharged. Therefore, the number of severe cases admitted to the hospital may be small compared to studies in other countries.

## 5. Conclusions

In conclusion, this study informed the clinical characteristics and prognosis of elderly patients with severe COVID-19 in Korea and assessed the risk factors for in-hospital mortality. Elderly patients with severe COVID-19 had higher APACHE II score, were frailer and had more underlying diseases than younger patients. Therefore, they commonly required vasopressors and invasive mechanical ventilation, and had a poor prognosis. As the COVID-19 pandemic continues, the number of elderly patients infected keeps increasing. Nevertheless, we believe that the results of this paper can help understand characteristics and predict the prognosis of elderly patients with severe COVID-19.

## Figures and Tables

**Figure 1 jpm-11-01258-f001:**
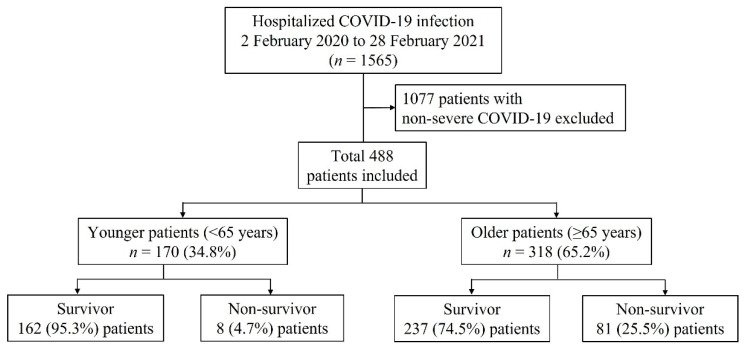
Patient flowchart, COVID-19: coronavirus disease 2019.

**Table 1 jpm-11-01258-t001:** Baseline characteristics of patients with severe COVID-19.

Variables	All Patients (*n* = 488)	Younger Patients (*n* = 170)	Older Patients (*n* = 318)	*p*-Value
Age	70 (62–80)	58 (51*–*62)	78 (71*–*84)	<0.001
Male (%)	234 (48.0)	98 (57.6)	136 (42.8)	0.002
Smoking (%)	87 (17.8)	30 (17.6)	57 (17.9)	0.939
Symptom at admission (%)	398 (81.6)	154 (90.6)	244 (76.7)	<0.001
Symptom to admission, days	4.0 (1.0*–*8.0)	5.0 (2.0*–*8.0)	3.0 (1.0*–*7.0)	0.860
Body mass index	24.3 (21.8*–*27.1)	26.0 (24.0*–*28.9)	23.1 (20.7*–*25.9)	<0.001
Resident of a nursing facilities	81 (16.6)	5 (2.9)	76 (23.9)	<0.001
Scoring systems				
APACHE II score	10.0 (7.0*–*13.3)	7.0 (5.0*–*10.0)	11.5 (9.0*–*15.0)	<0.001
Clinical Frailty Scale	3.0 (2.0*–*6.0)	2.0 (1.0*–*3.0)	4.0 (3.0*–*7.0)	<0.001
Charlson Comorbidity Index	3.0 (2.0*–*5.0)	2.0 (1.0*–*2.0)	4.0 (3.0*–*5.0)	<0.001
Comorbidity (%)				
Hypertension	266 (54.5)	65 (38.2)	201 (63.2)	<0.001
DM	150 (30.7)	34 (20.0)	116 (36.5)	<0.001
COPD	13 (2.7)	1 (0.6)	12 (3.8)	0.037
Cerebrovascular disease	49 (10.0)	7 (4.1)	42 (13.2)	0.001
Heart failure	17 (3.5)	1 (0.6)	16 (5.0)	0.011
Liver cirrhosis	5 (1.0)	1 (0.6)	4 (1.3)	0.484
Chronic kidney disease	5 (1.0)	0 (0)	5 (1.6)	0.100
Malignancy	27 (5.5)	9 (5.3)	18 (5.7)	0.866
Organ transplantation	1 (0.2)	1 (0.6)	0 (0)	0.171
Vital signs				
SBP, mmHg	121 (108*–*138)	122 (108*–*137)	121 (108*–*140)	0.906
DBP, mmHg	72 (63*–*83)	76 (66*–*86)	71 (61*–*81)	0.019
HR,/min	87 (75*–*100)	88 (80*–*102)	86 (73*–*99)	0.009
RR,/min	20 (20*–*22)	20 (20*–*22)	20 (19*–*24)	0.345
Body Temperature, ℃	36.8 (36.4*–*37.8)	37.4 (36.5*–*38.2)	36.7 (36.4*–*37.4)	<0.001
SpO_2_, %	96.0 (93.0*–*98.0)	96.0 (94.0*–*98.0)	95.0 (92.0*–*98.0)	0.129
GCS	15 (15*–*15)	15 (15*–*15)	15 (14*–*15)	0.002
Duration of fever	2.0 (0.0*–*5.0)	3.0 (1.0*–*5.3)	2.0 (0.0*–*4.3)	0.760
Laboratory data				
White cell count, 1000/mm^3^	5.87 (4.32*–*8.04)	5.77 (4.32*–*8.48)	5.91 (4.36*–*7.98)	0.767
Neutrophil-to-lymphocyte ratio	4.63 (2.71*–*8.88)	4.65 (2.62*–*7.79)	4.56 (2.75*–*9.60)	0.013
Hemoglobin, g/dL	12.9 (11.6*–*14.1)	13.8 (12.3*–*14.8)	12.6 (11.2*–*13.8)	<0.001
Platelet count, 1000/mm^3^	182 (136*–*229)	189 (150*–*237)	179 (134*–*228)	0.106
Total bilirubin, mg/dL	0.5 (0.4*–*0.7)	0.5 (0.4*–*0.8)	0.5 (0.3*–*0.7)	0.089
Albumin, g/dL	3.5 (3.1*–*4.0)	3.7 (3.3*–*4.2)	3.5 (3.1*–*3.8)	<0.001
BUN, mg/dL	15 (11*–*22)	13 (10*–*17)	16 (12*–*24)	0.003
Creatinine, mg/dL	0.76 (0.60*–*0.95)	0.73 (0.61*–*0.89)	0.78 (0.60*–*1.00)	0.992
C-reactive protein, mg/dL	5.93 (2.10*–*11.67)	6.40 (2.10*–*13.20)	5.70 (2.07*–*10.58)	0.322
Chest X-ray				
Normal	74 (15.2)	23 (13.5)	51 (16.0)	0.462
Unilateral	67 (13.7)	15 (8.8)	52 (16.4)	0.021
Bilateral	217 (44.5)	79 (46.5)	138 (43.4)	0.515
Multifocal	130 (26.6)	53 (31.2)	77 (24.2)	0.097

Data are presented as median and interquartile range or number (%), unless otherwise indicated. COVID-19: coronavirus disease 2019, APACHE II: Acute Physiology and Chronic Health Evaluation II, DM: diabetes mellitus, COPD: chronic obstructive pulmonary disease, SBP: systolic blood pressure, DBP: diastolic blood pressure, HR: heart rate, RR: respiratory rate, SpO_2_: saturation pulse oxygen, GCS: Glasgow Coma Scale, BUN: blood urea nitrogen.

**Table 2 jpm-11-01258-t002:** Treatment and clinical outcomes of patients with severe COVID-19.

Variables	All Patients (*n* = 488)	Younger Patients (*n* = 170)	Older Patients (*n* = 318)	*p*-Value
Treatment (%)				
Remdesivir	244 (50.0)	76 (44.7)	168 (52.8)	0.087
Antibiotics	294 (60.2)	83 (48.8)	211 (66.4)	<0.001
Vasopressor	61 (12.5)	13 (7.6)	48 (15.1)	0.018
CRRT	24 (4.9)	4 (2.4)	20 (6.3)	0.055
Corticosteroid	369 (75.6)	125 (73.5)	244 (76.7)	0.433
Oxygen supply device				
Nasal prong	408 (83.6)	151 (88.8)	257 (80.8)	0.023
HFNC	157 (32.2)	47 (27.6)	110 (34.6)	0.118
Invasive mechanical ventilation	120 (24.6)	40 (23.5)	80 (25.2)	0.691
ECMO	19 (3.9)	9 (5.3)	10 (3.1)	0.242
Tracheostomy (%)	37 (7.6)	10 (5.9)	27 (8.5)	0.300
Outcomes				
In-hospital mortality (%)	89 (18.2)	8 (4.7)	81 (25.5)	<0.001
Length of hospital stay (days)	16.0 (12.0*–*25.0)	16.0 (12.0*–*22.0)	17.0 (12.0*–*26.0)	0.396
Secondary infection	143 (29.3)	40 (23.5)	103 (32.4)	0.040
DNR order	81 (16.6)	5 (2.9)	76 (23.9)	<0.001

Data are presented as median and interquartile range or number (%), unless otherwise indicated. COVID-19: coronavirus disease 2019, CRRT: continuous renal replacement therapy, HFNC: high flow nasal cannula, ECMO: extracorporeal membrane oxygenation, DNR: do not resuscitate.

**Table 3 jpm-11-01258-t003:** Univariate and multivariate risk factors associated with in hospital mortality (Cox regression model).

	Univariate Analysis			Multivariate Analysis		
	HR	95% CI	*p*-Value	HR	95% CI	*p*-Value
Age	1.055	1.035–1.075	<0.001	1.031	1.008*–*1.054	0.008
Scoring systems						
APACHE II score	1.092	1.062*–*1.123	<0.001	1.038	1.006*–*1.071	0.018
Frailty scale	1.289	1.173*–*1.417	<0.001	1.022	0.887*–*1.178	0.764
Comorbidity (%)						
DM	2.115	1.391*–*3.214	<0.001	1.622	1.029*–*2.559	0.037
COPD	4.507	2.063*–*9.844	<0.001	4.294	1.874*–*9.837	0.001
Cerebrovascular disease	1.739	0.942*–*3.212	0.077	1.491	0.780*–*2.848	0.227
Liver cirrhosis	4.551	1.431*–*14.469	0.010	1.287	0.270*–*6.147	0.752
Laboratory findings						
White blood cell count, 1000/mm^3^	1.058	1.017*–*1.100	0.005	1.104	1.056*–*1.153	<0.001
Neutrophil lymphocyte ratio	1.020	1.007*–*1.032	0.002	0.984	0.968*–*1.000	0.044
Platelet count, 1000/mm^3^	0.994	0.991*–*0.998	0.001	0.996	0.993*–*0.999	0.024
Albumin, g/dL	0.451	0.312*–*0.652	<0.001	0.794	0.481*–*1.310	0.366
Creatinine, mg/dL	1.112	1.019*–*1.212	0.017	0.944	0.813*–*1.095	0.446
DNR order	9.618	6.102*–*15.159	<0.001	6.260	3.792*–*10.333	<0.001
Treatment						
Invasive mechanical ventilation	1.754	1.104*–*2.785	0.017	1.824	1.070*–*3.107	0.027
Vasopressor use	2.416	1.536*–*3.800	<0.001	1.069	0.583*–*1.961	0.829

HR: hazard ratio, CI: confidence interval, APACHE II: Acute Physiology and Chronic Health Evaluation II, DM: diabetes mellitus, COPD: chronic obstructive pulmonary disease, DNR: do not resuscitate.

## Data Availability

All data generated or analyzed during this study are included in this published article.

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
