# Peer review of "Comparison of Clinical Characteristics and Outcomes of Younger and Elderly Patients with Severe COVID-19 in Korea: A Retrospective Multicenter Study"

_jpm, 2021, doi:10.3390/jpm11121258_

Round 1

Reviewer 1 Report

The authors presented a work from multiple facilities from Korea based on a retrospective study. They have tried to presented clinical characteristics, outcomes, and factors associated with poor outcomes (death) among patients admitted with severe COVID-19. They also tried to make focused analysis among the elderly. The manuscript has an important scientific merits. However, the authors should address the following issues to make more interesting to the wider scientific community.

Major issues:

  1. Title: the manuscript presents clinical characteristics and outcomes of all patients admitted with severe COVID-19 (in Korea). I don’t understand why the authors preferred to present the manuscript as if only the elderly were studied. I agree with the authors that comparing outcome difference between the elderly and younger patients is logical and important. However, I suggest that authors to modify the title to align it with the overall content of the manuscript.

  2. Abstract: not separated to subheadings (background, result, and conclusion). This be done unless the journal wants such style. 
  3. Objective: The objective of the study was described as “We aimed to investigate to provide physicians a basis for treatment decision making”. However, this cannot be presented as an objective; it is rather the intended impact. Authors should rephrase the object as “to assess the clinical characteristics and outcome of patients hospitalized with severe COVID-19 to tertiary care centers in Republic of Korea
  4. Introduction: Introduction very short. Supplement it by including:
    (a) COVID-19 Korea republic (b) Why elderly people have more severe disease and high mortality from COVID-19 (c) predictors of mortality among COVID-19 patients.
    • Line 45 - 47 'Moreover, despite the use of remdesivir [2], steroids [3,4], and in- terleukin-6 antagonists [5-7] and vaccination [8-10], the incidence of COVID-19 has not reduced. This statement is not appropriate because (1) none of the items except vaccine are expected to reduce incidence of COVID-19 (2) the study was conducted before vaccine became widely available. The sentence should be replaced with sound statement.
  5. Methods: Supplement it with information such as how and why the seven hospitals were selected, how the participants were selected, if there were exclusion criteria used.
    • What is secondary infection? Do authors mean secondary bacterial infection. If that is the case, they have to explicitly describe how diagnosis of secondary bacterial infection (pneumonia, UTI, sepsis…) was made.

  6. Result: The result was presented redundantly. I understand that the authors wanted to compare the elderly against the younger patients as they also wanted to make subgroup analysis among elderly by comparing survivors against the non-survivors. The authors should make a choice between presenting all cases with severe disease or only the elderly. In that way, they can minimize reductant presentation of the result. The tables can also be reduced (to <=4) to make them easy for the readers.

Most of the contents of Table 1 were also described in the text in the main document. Authors should be selective and minimize duplicating findings both in the table and text.

Figure 1 – the manuscript presents outcome difference between the geriatric group and younger patients. However, the figure was presented as if the younger participants were excluded from outcome analysis. To align the figure with the contents of the manuscript, the younger patient group (<65) should dichotomize to survivors and non-survivors as it was done for older patients.

Minor comments:

  1. There are grammatical errors, long sentences, and problems with punctuation use in the manuscript. Authors should make correction of such errors to improve readability of the article. 
  2. Replace 'Non-survived' with 'non-survivor'

Author Response

Reviewer: 1

The authors presented a work from multiple facilities from Korea based on a retrospective study. They have tried to presented clinical characteristics, outcomes, and factors associated with poor outcomes (death) among patients admitted with severe COVID-19. They also tried to make focused analysis among the elderly. The manuscript has an important scientific merits. However, the authors should address the following issues to make more interesting to the wider scientific community.

Thank you for your comments. We did our best to answer all the questions and comments raised by reviewers.

Major issues:

Title: the manuscript presents clinical characteristics and outcomes of all patients admitted with severe COVID-19 (in Korea). I don’t understand why the authors preferred to present the manuscript as if only the elderly were studied. I agree with the authors that comparing outcome difference between the elderly and younger patients is logical and important. However, I suggest that authors to modify the title to align it with the overall content of the manuscript.

Authors’ Response : Thank you for your comments. As commented by reviewer, we revised the title as below.

Revised manuscript (Title)

Comparison of clinical characteristics and outcomes of younger and elderly patients with severe COVID-19 in Korea: a retrospective multicenter study

Abstract: not separated to subheadings (background, result, and conclusion). This be done unless the journal wants such style.

Authors’ Response : Thank you for your comments. As commented by reviewer, we are not separated to subheadings (background, result, and conclusion).

Objective: The objective of the study was described as “We aimed to investigate to provide physicians a basis for treatment decision making”. However, this cannot be presented as an objective; it is rather the intended impact. Authors should rephrase the object as “to assess the clinical characteristics and outcome of patients hospitalized with severe COVID-19 to tertiary care centers in Republic of Korea”

Authors’ Response : We are sorry to make confusion. We have changed a description of this in the objective area as follows.

Revised manuscript (Objective)

Old age is associated with disease severity and poor prognosis among coronavirus disease 2019 (COVID-19). However, characteristics of elderly patients with severe COVID-19 are limited. We aimed to assess the clinical characteristics and outcome of patients hospitalized with severe COVID-19 to tertiary care centers in South Korea; This retrospective multicenter study included patients with severe COVID-19 who were admitted at seven hospitals in South Korea from Feb-ruary 2, 2020, to February 28, 2021. The Cox regression analyses were performed to assess factors associated with the in-hospital mortality

Introduction: Introduction very short. Supplement it by including: (a) COVID-19 Korea republic (b) Why elderly people have more severe disease and high mortality from COVID-19 (c) predictors of mortality among COVID-19 patients.

Authors’ Response : Thank you for your comments. As commented by reviewer, we revised the introduction section as below.

Revised manuscript (Introduction)

Coronavirus disease 2019 (COVID-19) is primarily caused by severe acute respiratory syndrome coronavirus 2 (SARS-CoV-2), which is transmitted from person to person via close contact with an infected individual. A pandemic outbreak was declared by the World Health Organization (WHO) in March 2020 [1]. In South Korea, from 3 January 2020 to 15 November 2021, there have been 397,466 confirmed cases of COVID-19 with 3,115 deaths, reported to WHO. As of 6 November 2021, a total of 79,356,440 vaccine doses have been administered [2]. Vaccination [3-5] is currently in place worldwide and this is expected to reduce the incidence of COVID-19, but the world is still struggling with COVID-19.

The most common underlying diseases as predictors of mortality among COVID-19 patients were hypertension, diabetes, and cardiovascular diseases [6]. In addition, PaO2/FiO2 ratio (P/F ratio) ≤ 200 mmHg, respiratory failure at admission, high neutrophil and low lymphocyte, platelet, and albumin levels, and high sequential organ failure assessment (SOFA) score are known to be associated with mortality [7-9]. Old age is associated with greater disease severity and poor prognosis among patients with COVID-19 [7-10].

Elderly patients with COVID-19 patients present with more atypical symptoms and commonly experience progression to severe COVID-19 compared with younger patients [11, 12]. In addition, they have a higher in-hospital mortality rate and a longer length of hospital stay [13, 14]. The higher risk of COVID-19 mortality among older people occurred because older people were more likely to have other risk factors such as underlying disease and a high frailty scale. Also, older age is independently associated with COVID-19 mortality without other risk factors [15, 16]. However, data about the prognostic factors and characteristics of elderly patients with severe COVID-19 in a rapidly aging society are limited. Therefore, the current study aimed to assess the clinical characteristics and outcomes of patients hospitalized with severe COVID-19 to tertiary care centers in South Korea.

Line 45 - 47 'Moreover, despite the use of remdesivir [2], steroids [3,4], and in- terleukin-6 antagonists [5-7] and vaccination [8-10], the incidence of COVID-19 has not reduced. This statement is not appropriate because (1) none of the items except vaccine are expected to reduce incidence of COVID-19 (2) the study was conducted before vaccine became widely available. The sentence should be replaced with sound statement.

Authors’ Response : We are sorry to make confusion. No other treatment can reduce the incidence of COVID-19, and vaccination is now available worldwide. We have changed a description of this in the introduction area as follows.

Revised manuscript (Introduction)

Coronavirus disease 2019 (COVID-19) is primarily caused by severe acute respiratory syndrome coronavirus 2 (SARS-CoV-2), which is transmitted from person to person via close contact with an infected individual. A pandemic outbreak was declared by the World Health Organization (WHO) in March 2020 [1]. In South Korea, from 3 January 2020 to 15 November 2021, there have been 397,466 confirmed cases of COVID-19 with 3,115 deaths, reported to WHO. As of 6 November 2021, a total of 79,356,440 vaccine doses have been administered [2]. Vaccination [3-5] is currently in place worldwide and this is expected to reduce the incidence of COVID-19, but the world is still struggling with COVID-19.

Methods: Supplement it with information such as how and why the seven hospitals were selected, how the participants were selected, if there were exclusion criteria used.

Authors’ Response : Thank you for your comments. This is a multicenter, observational cohort study that enrolled adult (≥17 years old) patients with COVID-19 who were admitted to one of the ICUs at the 7 tertiary or referral hospitals in South Korea. Hospitals participating in this study contacted tertiary or referral hospitals that managed critically ill patients with COVID-19 in each state in South Korea. Patients not meeting the definition of severe COVID-19 were excluded in this study. During the study period, 1,565 patients were screened (Figure 1). A total of 1,077 patients were excluded due to non-severe COVID-19. As commented by reviewer, we revised the method section as below.

Revised manuscript (Method)

This multicenter retrospective study analyzed data collected from all patients with severe COVID-19 who were hospitalized at seven hospitals in South Korea from February 2, 2020, to February 28, 2021. The research was approved by the ethical committee of the promoting center (institutional review board of Chungnam National University Hospital, approval no. 2021-04-053) and by the local ethical committees. The need for a written informed consent was waived because of the retrospective nature of the study. During the study period, 1,565 patients were screened (Figure 1). A total of 1,077 patients were excluded due to non-severe COVID-19.

What is secondary infection? Do authors mean secondary bacterial infection. If that is the case, they have to explicitly describe how diagnosis of secondary bacterial infection (pneumonia, UTI, sepsis…) was made.

Authors’ Response : Thank you for your comments. We termed secondary infections to include both secondary bacterial and secondary fungal infections. Secondary infection (pneumonia, blood stream infection, urinary tract infection, etc) was defined as a pathogen identified in respiratory bacterial PCR (using endotracheal aspirates and expectorated sputum), nasopharyngeal PCR, blood culture and urine culture during illness or hospital stay and showed signs of infection. As commented by reviewer, we revised the method section as below.

Revised manuscript (Method)

Definition

The approval of requests was reserved for hospitalized patients who had SARS-CoV-2 infection confirmed via reverse-transcriptase polymerase-chain-reaction (PCR). Severe COVID-19 infection was defined as an oxygen saturation level of 94% or less while the patient was breathing ambient air and a need for oxygen support [17]. Secondary infection was defined as the presence of clinical signs and/or symptoms of infection and the presence of a pathogen based on the diagnostic tests, including respiratory bacterial PCR (using endotracheal aspirates and expectorated sputum), nasopharyngeal PCR, blood culture, and urine culture during illness or hospital stay [18].

Result: The result was presented redundantly. I understand that the authors wanted to compare the elderly against the younger patients as they also wanted to make subgroup analysis among elderly by comparing survivors against the non-survivors. The authors should make a choice between presenting all cases with severe disease or only the elderly. In that way, they can minimize reductant presentation of the result. The tables can also be reduced (to <=4) to make them easy for the readers.

Authors’ Response : Thank you for your comments. Our authors decided to describe the entire severe case. Therefore, we decided to express the contents more concisely by deleting the subgroup analysis of the elderly group. As commented by reviewer, we revised the result section as below.

Revised manuscript (Result)

Subgroup analysis of older patients with severe COVID-19 (≥ 65 years)

Among older patients with severe COVID-19, 237 (74.5%) survived and 81 (25.5%) non-survived (Figure 1). The non-survivor group was older and had higher APACHE II score, CCI, and CFS score. In terms of underlying diseases, DM and COPD were more common in the non-survivor group than in the survivor group. Moreover, the non-survivor group had a lower DBP and faster respiratory rate than the survivor group. In terms of laboratory data, non-survivor group had a higher WBC count, NLR, and BUN, creatinine, and c-reactive protein (CRP) level than the survivor group. However, non-survivor group had lower hemoglobin and albumin level and platelet count than the survivor group. The survivor group had a higher proportion of patients with normal initial chest radiographic findings (Table 4).

The use of antibiotics, vasopressor, CRRT, and corticosteroids were more frequent in the non-survivor group than in the survivor group. HFNC, invasive mechanical ventilation, extracorporeal membrane oxygenation (ECMO), and tracheostomy, but not the use of nasal prong, were more common in the non-survivor group than in the survivor group. The incidence of secondary infection and DNR order were more frequent in the non-survivor group than in the survivor group (Table 5).

The risk factor for in-hospital mortality among older patients was calculated using the Cox regression model (Table 6). After adjusting for confounders, the independent predictors of in-hospital mortality were age, presence of COPD, high WBC count, low NLR, albumin level, DNR order, and treatment with invasive mechanical ventilation.

Most of the contents of Table 1 were also described in the text in the main document. Authors should be selective and minimize duplicating findings both in the table and text.

Authors’ Response : We are sorry to make confusion. We try to selective and minimize duplicating findings both in the table and text. We changed a description of this in the result area as follows.

Revised manuscript (Result)

Table 1 shows the baseline characteristics of patients. The median ages of younger and older patient groups were 58 (IQR: 51–62) years and 78 (IQR: 71–84) years, respectively. The older patient group had a lower proportion of male patients and fewer symptoms upon admission. Further, the older patient group had a lower body mass index (BMI) and a higher proportion of nursing facility residents. Older patients had higher APACHE II scores (11.5 [9.0–15.0] vs. 7.0 [5.0–10.0], p < 0.001), CCI (4.0 [3.0–5.0] vs. 2.0 [1.0–2.0], p < 0.001), and CFS scores (4.0 [3.0–7.0] vs. 2.0 [1.0–3.0], p < 0.001) than younger patients. Hypertension, diabetes mellitus (DM), chronic obstructive pulmonary disease (COPD), cardiovascular disease, and heart failure were more common in older patients than in younger patients. In terms of vital signs, the older patients had a lower diastolic blood pressure (DBP), heart rate (HR), and body temperature than the younger patients. Regarding the initial laboratory data, compared with the younger patients, the older patients had lower neutrophil-to-lymphocyte ratios (NLRs), hemoglobin levels, and albumin level and higher blood urea nitrogen (BUN) levels. Based on the initial radiography result, unilateral lesions were more common in older patients than in younger patients.

Figure 1 – the manuscript presents outcome difference between the geriatric group and younger patients. However, the figure was presented as if the younger participants were excluded from outcome analysis. To align the figure with the contents of the manuscript, the younger patient group (<65) should dichotomize to survivors and non-survivors as it was done for older patients.

Authors’ Response : We are sorry to make confusion. We have added a description of this in the figure area as follows.

Revised manuscript (Result-Figure)

file submitted

Minor comments:

There are grammatical errors, long sentences, and problems with punctuation use in the manuscript. Authors should make correction of such errors to improve readability of the article.

Authors’ Response : Thank you for your comments. We have been corrected for grammatical errors, long sentences, and problems in enago. I will attach the proofreading document and submit it.

Replace 'Non-survived' with 'non-survivor'

Authors’ Response : Thank you for your comments. As commented by reviewer, we replace 'Non-survived' with 'non-survivor'

  1. Cucinotta D, Vanelli M: WHO Declares COVID-19 a Pandemic. Acta bio-medica : Atenei Parmensis 2020, 91(1):157-160.
  2. COVID-19, Korea [https://covid19.who.int/region/wpro/country/kr]
  3. Baden LR, El Sahly HM, Essink B, Kotloff K, Frey S, Novak R, Diemert D, Spector SA, Rouphael N, Creech CB et al: Efficacy and Safety of the mRNA-1273 SARS-CoV-2 Vaccine. The New England journal of medicine 2021, 384(5):403-416.
  4. Polack FP, Thomas SJ, Kitchin N, Absalon J, Gurtman A, Lockhart S, Perez JL, Pérez Marc G, Moreira ED, Zerbini C et al: Safety and Efficacy of the BNT162b2 mRNA Covid-19 Vaccine. The New England journal of medicine 2020, 383(27):2603-2615.
  5. Baden LR, El Sahly HM, Essink B, Kotloff K, Frey S, Novak R, Diemert D, Spector SA, Rouphael N, Creech CB et al: Efficacy and Safety of the mRNA-1273 SARS-CoV-2 Vaccine. The New England journal of medicine 2020.
  6. Javanmardi F, Keshavarzi A, Akbari A, Emami A, Pirbonyeh N: Prevalence of underlying diseases in died cases of COVID-19: A systematic review and meta-analysis. PloS one 2020, 15(10):e0241265.
  7. Booth A, Reed AB, Ponzo S, Yassaee A, Aral M, Plans D, Labrique A, Mohan D: Population risk factors for severe disease and mortality in COVID-19: A global systematic review and meta-analysis. PloS one 2021, 16(3):e0247461.
  8. Ruan Q, Yang K, Wang W, Jiang L, Song J: Clinical predictors of mortality due to COVID-19 based on an analysis of data of 150 patients from Wuhan, China. Intensive care medicine 2020, 46(5):846-848.
  9. Zhang L, Hou J, Ma FZ, Li J, Xue S, Xu ZG: The common risk factors for progression and mortality in COVID-19 patients: a meta-analysis. Archives of virology 2021:1-17.
  10. Huang C, Wang Y, Li X, Ren L, Zhao J, Hu Y, Zhang L, Fan G, Xu J, Gu X et al: Clinical features of patients infected with 2019 novel coronavirus in Wuhan, China. Lancet (London, England) 2020, 395(10223):497-506.
  11. Gan JM, Kho J, Akhunbay-Fudge M, Choo HM, Wright M, Batt F, Mandal AKJ, Chauhan R, Missouris CG: Atypical presentation of COVID-19 in hospitalised older adults. Ir J Med Sci 2021, 190(2):469-474.
  12. Bavaro DF, Diella L, Fabrizio C, Sulpasso R, Bottalico IF, Calamo A, Santoro CR, Brindicci G, Bruno G, Mastroianni A et al: Peculiar clinical presentation of COVID-19 and predictors of mortality in the elderly: A multicentre retrospective cohort study. International journal of infectious diseases : IJID : official publication of the International Society for Infectious Diseases 2021, 105:709-715.
  13. Malhotra V, Basu S, Sharma N, Kumar S, Garg S, Dushyant K, Borle A: Outcomes among 10,314 hospitalized COVID-19 patients at a tertiary care government hospital in Delhi, India. Journal of medical virology 2021, 93(7):4553-4558.
  14. Li X, Xu S, Yu M, Wang K, Tao Y, Zhou Y, Shi J, Zhou M, Wu B, Yang Z et al: Risk factors for severity and mortality in adult COVID-19 inpatients in Wuhan. The Journal of allergy and clinical immunology 2020, 146(1):110-118.
  15. Ho FK, Petermann-Rocha F, Gray SR, Jani BD, Katikireddi SV, Niedzwiedz CL, Foster H, Hastie CE, Mackay DF, Gill JMR et al: Is older age associated with COVID-19 mortality in the absence of other risk factors? General population cohort study of 470,034 participants. PloS one 2020, 15(11):e0241824.
  16. Owen RK, Conroy SP, Taub N, Jones W, Bryden D, Pareek M, Faull C, Abrams KR, Davis D, Banerjee J: Comparing associations between frailty and mortality in hospitalised older adults with or without COVID-19 infection: a retrospective observational study using electronic health records. Age Ageing 2021, 50(2):307-316.
  17. Berlin DA, Gulick RM, Martinez FJ: Severe Covid-19. The New England journal of medicine 2020, 383(25):2451-2460.
  18. Langford BJ, So M, Raybardhan S, Leung V, Westwood D, MacFadden DR, Soucy JR, Daneman N: Bacterial co-infection and secondary infection in patients with COVID-19: a living rapid review and meta-analysis. Clinical microbiology and infection : the official publication of the European Society of Clinical Microbiology and Infectious Diseases 2020.

Reviewer 2 Report

This Korean retrospective multicenter study describes the factors associated with the mortality of elderly patients with severe COVID 19.

The text is generally well written and clear, despite some unnecessary lengths at the start of the discussion, in particular

The contribution compared to the existing literature is moderate

Some major points deserve to be improved or clarified:

  • Lines 92-95: It is surprising to note that 2/3 of hospitalized patients are not severe. This does not correspond to the usual practice. It would therefore be necessary to clarify the justification for the frequency of his hospitalizations of non-severe cases.
  • The results text is too redundant with the tables, it would be possible to simplify the text to bring out the important points. Likewise, the first paragraph of the discussion should be more concise. You have already developed the results at the beginning of the following paragraphs.

I will add 2 minor remarks

  • In the introduction (line 32), the 2 percentages must be reversed
  • Line 45-47 The treatments cited are not intended to reduce the incidence, apart from vaccination. It is necessary to rephrase

Author Response

Reviewer: 2

Comments and Suggestions for Authors

This Korean retrospective multicenter study describes the factors associated with the mortality of elderly patients with severe COVID 19.

The text is generally well written and clear, despite some unnecessary lengths at the start of the discussion, in particular

The contribution compared to the existing literature is moderate

Some major points deserve to be improved or clarified:

Thank you for your comments. We did our best to answer all the questions and comments raised by reviewers.

Lines 92-95: It is surprising to note that 2/3 of hospitalized patients are not severe. This does not correspond to the usual practice. It would therefore be necessary to clarify the justification for the frequency of his hospitalizations of non-severe cases.

Authors’ Response : Thank you for your comments. In South Korea, at the beginning of the COVID-19 outbreak, regardless of severity, all patients were hospitalized and then discharged. Therefore, non-severe cases may be higher than in other countries. As commented by reviewer, we revised the limitation section as below.

Revised manuscript (Limitation)

This study had several limitations. First, it included patients admitted at tertiary or referral hospitals capable of critical care. This might have affected the results, as patients who had been transferred from other hospitals or from living treatment centers due to worsening conditions were included. Second, instead of manually reviewing medical records, data were collected from the electronic health record database. Hence, some details could have been missing. Third, the patient's economic status was not assessed even though inferior economic conditions are associated with poor outcomes [19, 20]. Nevertheless, in Korea, the National Health Insurance provides free COVID-19 treatment. Hence the impact is likely low. Fourth, In South Korea, at the beginning of the COVID-19 outbreak, regardless of severity, all patients were hospitalized and then discharged. Therefore, the number of severe cases admitted to the hospital may be small compared to studies in other countries.

The results text is too redundant with the tables, it would be possible to simplify the text to bring out the important points. Likewise, the first paragraph of the discussion should be more concise. You have already developed the results at the beginning of the following paragraphs.

Authors’ Response : Thank you for your comments. We try to simplify the table and first paragraph of the discussion. As commented by reviewer, we revised the result and discussion section as below.

Revised manuscript (Result)

Table 1 shows the baseline characteristics of patients. The median ages of younger and older patient groups were 58 (IQR: 51–62) years and 78 (IQR: 71–84) years, respectively. The older patient group had a lower proportion of male patients and fewer symptoms upon admission. Further, the older patient group had a lower body mass index (BMI) and a higher proportion of nursing facility residents. Older patients had higher APACHE II scores (11.5 [9.0–15.0] vs. 7.0 [5.0–10.0], p < 0.001), CCI (4.0 [3.0–5.0] vs. 2.0 [1.0–2.0], p < 0.001), and CFS scores (4.0 [3.0–7.0] vs. 2.0 [1.0–3.0], p < 0.001) than younger patients. Hypertension, diabetes mellitus (DM), chronic obstructive pulmonary disease (COPD), cardiovascular disease, and heart failure were more common in older patients than in younger patients. In terms of vital signs, the older patients had a lower diastolic blood pressure (DBP), heart rate (HR), and body temperature than the younger patients. Regarding the initial laboratory data, compared with the younger patients, the older patients had lower neutrophil-to-lymphocyte ratios (NLRs), hemoglobin levels, and albumin level and higher blood urea nitrogen (BUN) levels. Based on the initial radiography result, unilateral lesions were more common in older patients than in younger patients.

Revised manuscript (Discussion)

This multicenter study investigated the correlation between in-hospital mortality and some factors among elderly patients with severe COVID-19. Approximately 65.2% of hospitalized patients with severe COVID-19 were aged over 65 years. The older patient group had a lower BMI and higher proportion of nursing facility residents.  Older patients had higher initial APACHE II scores, CCI, and CFS scores than younger patients. Moreover, they commonly presented with underlying diseases, lower NLRs and hemoglobin and albumin levels, and higher BUN levels. The use of antibiotics and vasopressors was higher in older patients than in younger patients. Older patients had a higher in-hospital mortality rate and incidence of secondary infection than younger patients, and DNR order were more common in older patients than in younger patients. The risk factors for in-hospital mortality in patients with severe COVID-19 included high APACHE II score and WBC count, low platelet count, DNR order, and treatment with invasive mechanical ventilation. The in-hospital mortality rate of the older patient group was 25.5%. The non-survivor group had higher severity and CFS scores than the survivor group.  The risk factors for in-hospital mortality among older patients were age, presence of COPD, high WBC count, low NLR, low albumin level, DNR order, and treatment with invasive mechanical ventilation.

I will add 2 minor remarks

In the introduction (line 32), the 2 percentages must be reversed

Authors’ Response : We are sorry to make confusion. As commented by reviewer, we revised the abstract section as below.

Revised manuscript (Introduction)

The in-hospital mortality rate was 18.2%, and the older patient group had a higher in-hospital mortality rate than the younger patient group (25.5% vs 4.7%, p-value < 0.001).

Line 45-47 The treatments cited are not intended to reduce the incidence, apart from vaccination. It is necessary to rephrase

Authors’ Response : Thank you for your comments. As commented by reviewer, we revised the introduction section as below.

Revised manuscript (Introduction)

Coronavirus disease 2019 (COVID-19) is primarily caused by severe acute respiratory syndrome coronavirus 2 (SARS-CoV-2), which is transmitted from person to person via close contact with an infected individual. A pandemic outbreak was declared by the World Health Organization (WHO) in March 2020 [1]. In South Korea, from 3 January 2020 to 4:30pm central European time, 15 November 2021, there have been 397,466 confirmed cases of COVID-19 with 3,115 deaths, reported to WHO. As of 6 November 2021, a total of 79,356,440 vaccine doses have been administered [2]. Vaccination [3-5] is currently in place worldwide and this is expected to reduce the incidence of COVID-19, but the world is still struggling with COVID-19.

  1. Cucinotta D, Vanelli M: WHO Declares COVID-19 a Pandemic. Acta bio-medica : Atenei Parmensis 2020, 91(1):157-160.
  2. COVID-19, Korea [https://covid19.who.int/region/wpro/country/kr]
  3. Baden LR, El Sahly HM, Essink B, Kotloff K, Frey S, Novak R, Diemert D, Spector SA, Rouphael N, Creech CB et al: Efficacy and Safety of the mRNA-1273 SARS-CoV-2 Vaccine. The New England journal of medicine 2021, 384(5):403-416.
  4. Polack FP, Thomas SJ, Kitchin N, Absalon J, Gurtman A, Lockhart S, Perez JL, Pérez Marc G, Moreira ED, Zerbini C et al: Safety and Efficacy of the BNT162b2 mRNA Covid-19 Vaccine. The New England journal of medicine 2020, 383(27):2603-2615.
  5. Baden LR, El Sahly HM, Essink B, Kotloff K, Frey S, Novak R, Diemert D, Spector SA, Rouphael N, Creech CB et al: Efficacy and Safety of the mRNA-1273 SARS-CoV-2 Vaccine. The New England journal of medicine 2020.
  1. Wernly B, Beil M, Bruno RR, Binnebössel S, Kelm M, Sigal S, van Heerden PV, Boumendil A, Artigas A, Cecconi M et al: Provision of critical care for the elderly in Europe: a retrospective comparison of national healthcare frameworks in intensive care units. BMJ open 2021, 11(6):e046909.
  2. Baldwin MR, Sell JL, Heyden N, Javaid A, Berlin DA, Gonzalez WC, Bach PB, Maurer MS, Lovasi GS, Lederer DJ: Race, Ethnicity, Health Insurance, and Mortality in Older Survivors of Critical Illness. Critical care medicine 2017, 45(6):e583-e591.

Round 2

Reviewer 1 Report

The authors have adequately addressed my comments

Author Response

The authors have adequately addressed my comments.

Thank you for your comments. We did our best to answer all the questions and comments raised by reviewers.

Reviewer 2 Report

The text has been significantly improved and simplified in the presentation of the results.

However, I find that the conclusion still lacks clarity. In particular line 311, the sentence should be rephrased "Elderly patients with severe COVID-19 had higher disease severity and frailer and had more underlying diseases."

The last sentence of the conclusion does not add much and is questionable. The treatment of a patient cannot be predicted from this study.

Author Response

The text has been significantly improved and simplified in the presentation of the results.

Thank you for your comments. We did our best to answer all the questions and comments raised by reviewers.

However, I find that the conclusion still lacks clarity. In particular line 311, the sentence should be rephrased "Elderly patients with severe COVID-19 had higher disease severity and frailer and had more underlying diseases."

Authors’ Response : We are sorry to make confusion. As commented by reviewer, we revised the conclusion section as below.

Revised manuscript (Conclusion)

In conclusion, this study informed the clinical characteristics and prognosis of elderly patients with severe COVID-19 in Korea and assessed the risk factors for in-hospital mortality. Elderly patients with severe COVID-19 had higher APACHE II score and frailer and had more underlying diseases than younger patients.

The last sentence of the conclusion does not add much and is questionable. The treatment of a patient cannot be predicted from this study.

Authors’ Response : Thank you for your comments. Our authors try to simplify and reduce the questionable sentence in the conclusion. As commented by reviewer, we revised the conclusion section as below.

Revised manuscript (Conclusion)

In conclusion, this study informed the clinical characteristics and prognosis of elderly patients with severe COVID-19 in Korea and assessed the risk factors for in-hospital mortality. Elderly patients with severe COVID-19 had higher APACHE II score and frailer and had more underlying diseases than younger patients. Therefore, they commonly required vasopressors and invasive mechanical ventilation and they had a poor prognosis. The risk factors of in-hospital mortality were age; disease severity upon admission; presence of DM, COPD, and leukocytosis; lower NLR and albumin level; DNR order, and treatment with invasive mechanical ventilation. As the COVID-19 pandemic continues, the number of elderly patients infected keeps increasing. Nevertheless, we believe that the results of this paper can help understand characteristics and predict the prognosis and treatment of elderly patients with severe COVID-19.
